# Issues with Value-Based Multi-objective Reinforcement Learning: Value Function Interference and Overestimation Sensitivity

## Abstract

Multi-objective reinforcement learning (MORL) algorithms extend conventional reinforcement learning (RL) to the more general case of problems with multiple, conflicting objectives, represented by vector-valued rewards. Widely-used scalar RL methods such as Q-learning can be modified to handle multiple objectives by (1) learning vector-valued value functions, and (2) performing action selection using a scalarisation or ordering operator which reflects the user's preferences with respect to the different objectives. This paper investigates two previously unreported issues which can hinder the performance of value-based MORL algorithms when applied in conjunction with a non-linear utility function – value function interference, and sensitivity to overestimation. We illustrate the nature of these phenomena on simple multi-objective MDPs using a tabular implementation of multiobjective Q-learning.

## 1 Introduction and Background

Multi-objective reinforcement learning (MORL) is a form of reinforcement learning (RL) designed for problems with multiple, conflicting objectives (Roijers et al., 2013; Hayes et al., 2022). Whereas conventional, single-objective RL algorithms assume that the agent is provided with a scalar reward after each action, in MORL these rewards are vectors with a separate component for each objective. Prior research has shown that applying single-objective methods to multi-objective tasks may result in unsuitable, inferior or inflexible solutions (Roijers et al., 2013; 2015), and so in recent years there has been considerable interest in developing MORL methods. In many cases these MORL algorithms are extensions of existing RL methods, including value-based approaches such as Q-learning (Gábor et al., 1998; Natarajan & Tadepalli, 2005; Van Moffaert et al., 2013; Van Moffaert & Nowé, 2014; Mossalam et al., 2016; Vamplew et al., 2017b; Abels et al., 2019).

Adapting value-based reinforcement learning methods to work with multiple objectives requires two key changes to be made:

1. As the rewards received by the agent are vectors, the Q-values stored by the agent must also be vectors.

2. When action selection is performed, a greedy action must be identified in a way which is consistent with the user's overall utility (Roijers et al., 2013; Zintgraf et al., 2015).

The action selection can in most cases be implemented via applying a *scalarisation* operator which maps the vector-values to scalars, and then regarding the action with the highest scalarised value as being the greedy selection. Depending on the user's utility preferences this might be a simple linear weighting or a non-linear operator, such as the Chebyshev distance. For some forms of utility (e.g. those based on lexicographic ordering, which often arise in the context of meeting constraints (Gábor et al., 1998) or in problems where fairness is a consideration (Siddique et al., 2020)), it may not be possible to represent the utility directly in terms of scalarisation (Debreu, 1997). In such cases, the Q-learning agent can instead use an ordering

operator to identify the greedy action. For simplicity in this paper we will restrict our examples to utilities which can be implemented via scalarisation.

Algorithm 1 provides a generic form of multi-objective Q-learning. The algorithm presented here is similar to that previously described in Vamplew et al. (2021a) and Ding et al. (2025), but has been modified to correctly handle discounting of future rewards. The agent maintains a vector $P$ of the discounted return accumulated so far (Line 14). This is combined with appropriately discounted Q-values for the current state to estimate the complete discounted return $V$ (Line 16), which is subsequently used in conjunction with $U$ to perform action selection (Lines 17-18).

Note that when a non-linear $U$ is used, then both the action selection and the Q-values learned by the agent must be conditioned not just on the current state of the environment, but also on some measure of the rewards received so far (Vamplew et al., 2021a).[1] This is the purpose of the *augmented state* created on Lines 10 and 15. Again, the algorithm specified here introduces a modification from that reported in the previous literature, in that the augmented state concatenates the environment state information with both the accumulated return vector $P$, and also the time-step. The need for the latter term has not to our knowledge previously been identified in the literature on multi-objective Q-learning – while Cai et al. (2023) do include $t$ within their definition of state (see footnote 3 on page 3 of their paper), they state that this is required only for the finite horizon setting. However it can be seen from Algorithm 1 that the value of $V$, and therefore the choice of action, depends on $t$ as this is used to determine the amount of discounting applied to the Q-values. It is possible that the structure of a MOMDP may allow for the agent to reach a particular environmental state $s$ with the same accumulated return $P$ after different numbers of time-steps. In this case the optimal policy may differ depending on the time-step, and hence this must be included in the augmented state $S^A$. Without this the policy may be non-stationary with respect to $S^A$ and the agent may fail to converge to a stable policy. In practice as the value of $t$ is potentially unbounded, it may be preferable to use $\gamma^t$ in the augmented state as this will naturally be normalised in the range $[0 \dots 1]$.

One further consideration in MORL is the choice of optimisation criteria, as there are two distinct options compared with just a single possible criteria in conventional RL (Roijers et al., 2013). The first one is to maximise the Expected Scalarised Return (ESR). In this approach, the agent aims to maximise the expected value which is first scalarised by the utility function, as shown below (Eq 1) where $w$ is the parameter vector for utility function $U$, $r_k$ is the vector reward on time-step $k$, and $\gamma$ is the discounting factor

$$V_{\mathbf{w}}^{\pi}(s) = E[U(\sum_{k=0}^{\infty} \gamma^k \mathbf{r}_k, \mathbf{w}) \mid \pi, s_0 = s] \tag{1}$$

ESR is the appropriate criteria for problems where we care about the outcome within each individual episode. For example, consider selecting a treatment plan for a patient, where there is a trade-off between the effectiveness of the treatment and negative side-effects. Each patient would care about their own individual outcome rather than the mean outcome over all patients.

The second criteria is the Scalarised Expected Return (SER) which estimates the expected vector return per episode and then maximises the scalarised expected return, as shown in (Eq 2)

$$V_{\mathbf{w}}^{\pi}(s) = U(\mathbf{V}^{\pi}(s), \mathbf{w}) = U(E[\sum_{k=0}^{\infty} \gamma^k \mathbf{r}_k \mid \pi, s_0 = s], \mathbf{w}) \tag{2}$$

SER is an appropriate criteria when we care about the optimal utility over multiple executions of a policy. For example a logistics company might seek to optimise both delivery time and fuel consumption, but as they make many deliveries they care about the average performance across all episodes rather than the outcome from each individual episode.

One further distinction between MORL algorithms is whether they are *single-policy* or *multi-policy* in nature (Vamplew et al., 2011; Roijers et al., 2013). The former aims to learn a single policy which is optimal for a

---

[1]An exception is in the case where rewards are zero except at terminal states. (Issabekov & Vamplew, 2012). For clarity the environments used in this paper do have this simplifying property. However the observations made here are equally applicable to the more general case.

---

**Algorithm 1** A general algorithm for multi-objective Q($\lambda$).

---

input: learning rate $\alpha$, discounting term $\gamma$, eligibility trace decay term $\lambda$, number of objectives $n$, scalarisation function $U$ and any associated parameters

1: **for** all augmented states $s^A$, actions $a$ and objectives $o$ **do**
2:     initialise $Q_o(s^A, a)$
3: **end for**
4: **for** each episode **do**
5:     **for** all augmented states $s^A$ and actions $a$ **do**
6:         $e(s^A, a) = 0$
7:     **end for**
8:     initialise sum of prior rewards $P$ to a zero vector
9:     observe initial state $s_0$
10:     $s_0^A = (s_0, P, \gamma^0)$                                              ▷ create augmented state
11:     select $a_0$ from an exploratory policy derived using $U(Q(s_0^A))$
12:     **for** each step $t$ of the episode **do**
13:         execute $a_t$, observe $s_{t+1}$ and reward $R_t$
14:         $P = P + \gamma^t R_t$
15:         $s_{t+1}^A = (s_{t+1}, P, \gamma^{t+1})$                         ▷ create augmented state
16:         $V(s_{t+1}^A) = P + \gamma^{t+1}Q(s_{t+1}^A)$              ▷ create value vector for all actions
17:         select $a^*$ from a greedy policy derived using $U(V(s_{t+1}^A))$
18:         select $a'$ from an exploratory policy derived using $U(V(s_{t+1}^A))$
19:         $\delta = R_t + \gamma Q(s_{t+1}^A, a^*) - Q(s_t^A, a_t)$
20:         $e(s_t^A, a_t) = 1$
21:         **for** each augmented state $s^A$ and action $a$ **do**
22:             $Q(s^A, a) = Q(s^A, a) + \alpha\delta e(s^A, a)$
23:             **if** $a' = a^*$ **then**
24:                 $e(s^A, a) = \gamma\lambda e(s^A, a)$
25:             **else**
26:                 $e(s^A, a) = 0$
27:             **end if**
28:         **end for**
29:         $a_{t+1} = a'$
30:     **end for**
31: **end for**

---

specific choice of utility parameters, whereas the latter learns a set of policies where each policy is optimal for a different choice of parameters for the utility function. For clarity this paper restricts its focus to single-policy algorithms, but the issues identified here would also arise in value-based multi-policy algorithms.

Multi-objective extensions of Q-learning have been amongst the most widely used forms of MORL reported so far in the literature (Gábor et al., 1998; Natarajan & Tadepalli, 2005; Van Moffaert et al., 2013; Van Moffaert & Nowé, 2014; Ruiz-Montiel et al., 2017; Vamplew et al., 2017b). However, as we will demonstrate via example and empirically in this paper, when used in combination with a non-linear utility function two issues can arise which may significantly slow learning or cause the agent to converge to a sub-optimal policy. Section 2 examines an issue which we label *value function interference* where the expected vector-values learned by the agent may lead to sub-optimal action selection. We show that this problem can occur in both stochastic and deterministic environments, and examine a modified form of action-selection which can mitigate (though not eliminate) the problem in the latter case. Section 3 examines the impact of *overestimation of Q-values* on MORL. It is well known from prior studies of scalar RL that methods such as Q-learning can be prone to overestimation (Thrun & Schwartz, 1993; Van Hasselt et al., 2016). Here we identify for the first time that this issue may be even more impactful on learning in the context of multiple objectives.

## 2 Interference in multi-objective value functions

As discussed in the Introduction, extensions of value-based RL methods such as Q-learning to MORL require learning vectors for the Q-values. In this type of algorithm, the vector-values learned for a particular state-action pair represent the expected return averaged over multiple different future returns (due to stochasticity in either the environment and/or the policy of the agent). When used in combination with a non-linear utility function $U$, the utility of this expected return vector may differ from that of the returns from which it is derived, and this may lead to action-selection which is not optimal with regards to $U$. We label this phenomenon *value function interference.*

The graphs in Figure 1 illustrate a set of circumstances under which value function interference may arise and cause learning of a sub-optimal policy. For simplicity of visualisation, this figure illustrates the case of a single-objective reward. While the application of a utility function is not standard practice in single-objective RL, it is feasible and potentially beneficial to do so (Vamplew et al., 2024). In both graphs, we assume the existence of two actions $a_1$ and $a_2$. We assume that $a_1$ stochastically produces two outcomes, with returns denoted $Q(a_1)^{low}$ and $Q(a_1)^{high}$, and that these occur with equal probability such that $Q(a_1) = (Q(a_1)^{low} + Q(a_1)^{high})/2$. Meanwhile $a_2$ is assumed to produce a deterministic outcome[2] denoted $Q(a_2)$.

As $a_2$ has deterministic outcomes, its utility will be the same under both the SER and ESR criterion, and equal to $U(Q(a_2))$. However as $a_1$ outcomes are stochastic, its utility will differ under these criterion. For SER, the utility is $U(Q(a_1))$ which equals $U(Q(a_1)^{low} * 0.5 + Q(a_1)^{high} * 0.5)$ (i.e. the estimated per-objective returns are probabilistically combined over all outcomes, and then the utility function is applied. This is the value which would be learned and used within Algorithm 1. In contrast, for ESR the relevant utility is $U(Q(a_1)^{low}) * 0.5 + U(Q(a_1)^{high}) * 0.5$ (i.e. a probabilistically weighted average of the scalarised return of each outcome). For non-linear definitions of $U$, these values will not be equal. Figure 1 shows that for a convex utility function (left) $SER(a_1) < ESR(a_1)$ whereas for a concave utility function (right) $ESR(a_1) < SER(a_1)$.

This issue may lead to selection of actions which are incompatible with optimising the ESR criterion, if another action ($a_2$) has an expected return falling within the shaded region in these graphs. In the case on the left, if $SER(a_1) < U(Q(a_2)) < ESR(a_1)$ then $a_2$ will incorrectly be identified as the optimal action whereas $a_1$ is preferable under ESR. Conversely in the case on the right, if $ESR(a_1) < U(Q(a_2)) < SER(a_1)$ then $a_1$ will incorrectly be identified as optimal action whereas $a_2$ is preferable under ESR.

---

[2]We note that value function interference may still arise if the outcomes of $a_2$ are stochastic, as long as the range of variation is lower than for $a_1$.

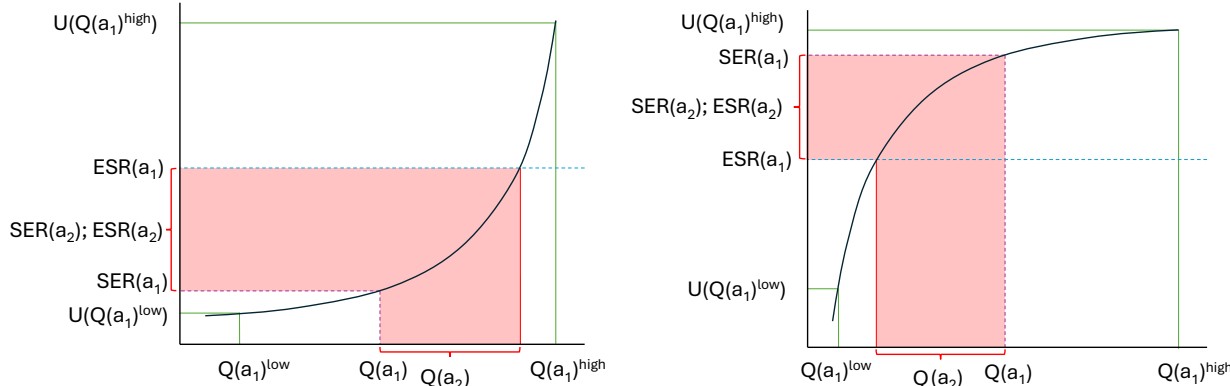

Figure 1: An illustration of value-function interference for both a convex and concave utility function. In both cases action $a_1$ has a stochastic return where $Q(a_1)^{low}$ and $Q(a_1)^{high}$ occur with equal probability. The highlighted region indicates conditions under which the agent may prefer the incorrect action with regards to ESR optimality.

## 2.1 Value function interference in stochastic MOMDPs

We start by considering environments with some element of stochasticity (in either the state transitions or rewards), as this is perhaps the more obvious context in which value function interference can occur. For stochastic environments the SER and ESR criteria are not equivalent, and value interference issues may arise for ESR-optimal learning which do not occur for SER-optimal learning.

Consider the very simple multi-objective Markov Decision Process (MOMDP)[3] shown in Figure 2. There are many ways in which a utility function could be derived from the rewards in this MOMDP, but let us assume that $U$ equals twice the value of the first objective minus the product of the second and third objectives (which are always negative)[4]. Action $a_1$ has a stochastic reward which varies between (7, -1, -5) and (7, -5, -1) with equal probability. The utility of this action differs depending on whether we are optimising with respect to ESR or SER. $U_{SER}(a_1)=5$ (the scalarised value of the expected vector reward) while $U_{ESR}(a_1)=9$ (the expected scalarised value, which is 9 for both of the possible outcomes). Action $a_2$ deterministically returns a reward of (8, -3, -3) – as this is deterministic $U_{SER}(a_2) = U_{ESR}(a_2) = 7$.

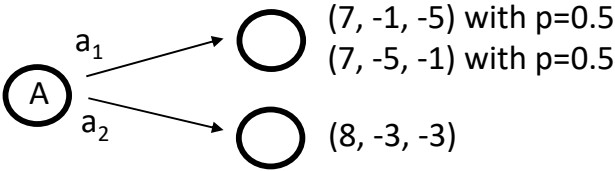

Figure 2: An example multi-objective MDP with stochastic rewards. The unlabelled states are terminal states, and the vector reward for reaching these states is indicated to their right.

Therefore the SER-optimal policy in this case is to select $a_2$, while the ESR-optimal policy selects $a_1$. The SER-optimal policy can be determined on the basis of the vector Q-values learned by the agent, but this is not true for the ESR-optimal policy, as these values only represent the expected vector return which is insufficient for the agent to determine the per-episode expected scalarised return.

---

[3]In fact, as this MOMDP has only a single non-terminal state, this example demonstrates that value function interference can arise even for ESR optimisation of multi-objective multi-armed bandits (MOMABs).

[4]The actual specification of rewards and utilities is outside the scope of this project; we focus here simply on examining whether MORL methods will optimise for the utilities and rewards as defined by the user, irrespective of the structure of those definitions. The issues identified here could arise for any non-linear definition of utility.

To demonstrate the impact of value function interference, experiments were run on this MOMAB using a tabular multi-objective Q-learning agent. A hyperparameter sweep was performed across different values for the learning rate ($\alpha$) and starting exploration parameter (using $\epsilon$-greedy exploration with $\epsilon$ linearly decayed to zero over the trial). The $\lambda$ and $\gamma$ hyperparameters were fixed at 0.95 and 1 respectively, and to produce more directed exploration Q-values were optimistically initialised to (12,0,0). For each set of hyperparameters 100 trials of each agent were executed. Each trial lasted for 500 episodes, and at the end of the trial the final greedy policy of the agent was identified. Figure 3 shows the final greedy arm selection over 100 trials of each hyperparameter setting. It clearly highlights the problems caused by value function interference, with all but one trial using a learning rate $\alpha \leq 0.7$ converging to the incorrect arm, thereby causing a large loss in utility. Only trials using a very high learning rate were successful, and such high learning rates are unlikely to work effectively for more complex stochastic MOMDPs.

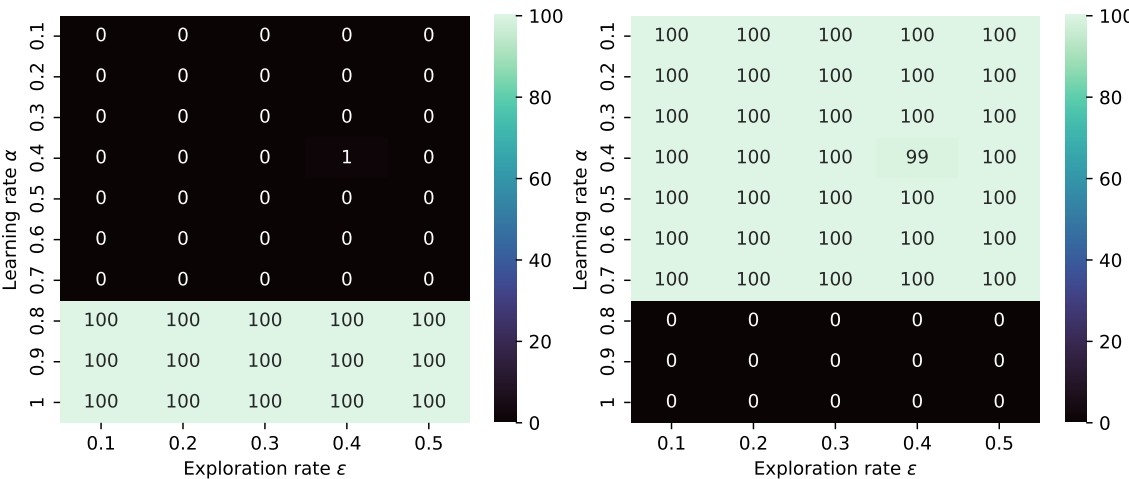

Figure 3: Heatmaps showing the frequency with which the correct arm $a_1$ (left) and incorrect arm $a_2$ (right) were selected as the final greedy arm for the stochastic MOMAB across 100 trials using different hyperparameter settings.

## 2.2 Value function interference in deterministic MOMDPs

Now consider the case of environments where the choice of starting state, the state transitions and the rewards are fully deterministic. Assuming that the policy being executed is also deterministic, then in this context the SER and ESR optimisation criteria are equivalent (every execution of a policy results in the same return, which of course is also the same as the mean return considered by SER). Given the fully deterministic characteristics of this setting, it is less clear how value function interference might arise. However we will demonstrate that it can still occur, due to two different underlying causes.

Value function interference manifests differently in the deterministic case. Rather than being evident in the values learned for the state-action pair where stochastic rewards occur as in Section 2.1, in this context issues arise with the values learned for predecessor states. Therefore rather than a MOMAB, we will require a multi-state MOMDP to demonstrate this issue. As an illustrative example, consider the simple deterministic MOMDP shown in Figure 4. This MOMDP consists of three states ($A$, $B$, and $C$). In each state two actions are available ($a_1$ and $a_2$). The MOMDP has three objectives, so the reward vector has three elements. There are four possible deterministic stationary policies, as summarised in Table 1.

Assuming the same utility function as used previously, $U(B, a_1) = U(B, a_2) = 9, U(C, a_1) = 7$, and $U(C, a_2) = -25$. Therefore clearly the optimal policy should be to select action $a_1$ in state $A$, followed by either action in state $B$ i.e. either policy 0 or policy 1 (this is true for both the SER and ESR criterion).

However the existence of two equally optimal actions in state $B$ creates an issue in the context of multi-objective rewards. The usual approach taken in Q-learning where more than one optimal action exists is to

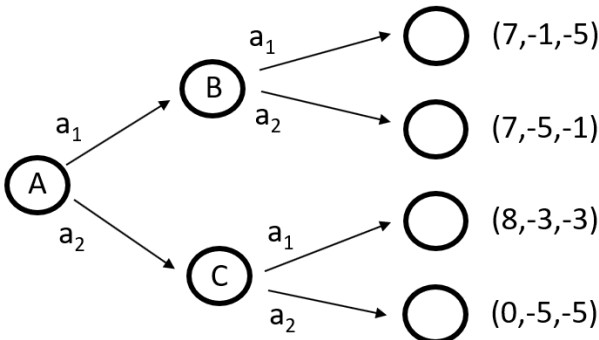

Figure 4: An example multi-objective MDP. The unlabelled states are terminal states, and the vector reward for reaching these states is indicated to their right. All other state transitions receive zero rewards.

| Policy label | Action in State A | Action in State B | Action in State C | Vector return | Utility |
|---|---|---|---|---|---|
| 0 | $a_1$ | $a_1$ | - | (7, -1, -5) | 9 |
| 1 | $a_1$ | $a_2$ | - | (7, -5, -1) | 9 |
| 2 | $a_2$ | - | $a_1$ | (8, -3, -3) | 7 |
| 3 | $a_2$ | - | $a_2$ | (0, -5, -5) | -25 |

Table 1: The four policies for the deterministic MOMDP from Figure 4. The scalar utility in the final column assumes that the utility equals twice the first objective minus the product of the second and third objectives.

simply randomly select between them. However this random selection violates our earlier assumption that the policy is deterministic. This introduces an element of stochasticity into the vector rewards which the agent receives, which in turn leads to interference in the values which the agent learns. This interference problem does not arise in single-objective RL, because two actions which are equally optimal must, by definition, share the same mean scalar reward, whereas in the multi-objective case two actions may share the same utility while having differing mean vector rewards.

Consider the values learned for $Q(A, a_1)$. This action always leads to state $B$, but at that stage under the optimal policy either action is equally likely to be selected, and so $Q(A, a_1)$ will converge to the mean of $Q(B, a_1)$ and $Q(B, a_2) = (7, -3, -3)$, which has an estimated utility of just 5. Meanwhile because action $a_2$ in state $A$ will always be followed by action $a_1$ in state $C$, $Q(A, a_2)$ will converge to the same value as $Q(C, a_1)$, and so will have utility of 7. Therefore ultimately the agent will converge incorrectly to the sub-optimal policy 2 (selecting action $a_2$ in state $A$ and $a_1$ in state $C$), with a considerable consequent loss of utility.

This example illustrates one of the ways that value function interference may arise within a deterministic environment. If the utility function allows actions with widely differing vector outcomes to map to the same utility value, then the vector value function learned for earlier states may be inconsistent with the actual optimal policy. We note that if the utility function is linear then value interference does not impact on selecting the optimal action. The value learned for $Q(A, a_1)$ will converge towards a point on the line segment joining $Q(B, a_1)$ and $Q(B, a_2)$. If $Q(B, a_1)$ and $Q(B, a_2)$ share the same utility value for a particular linear weighting of objectives, then all points on that line also have that same utility.

### 2.3 Empirical evaluation of the effect of value-function interference in a deterministic MOMDP

The problem of value function interference in a deterministic MOMDP can potentially be addressed by modifying the selection process so that ties between multiple greedy actions are broken in a deterministic fashion. For example, we might always select the greedy action with the lowest sub-script - this introduces

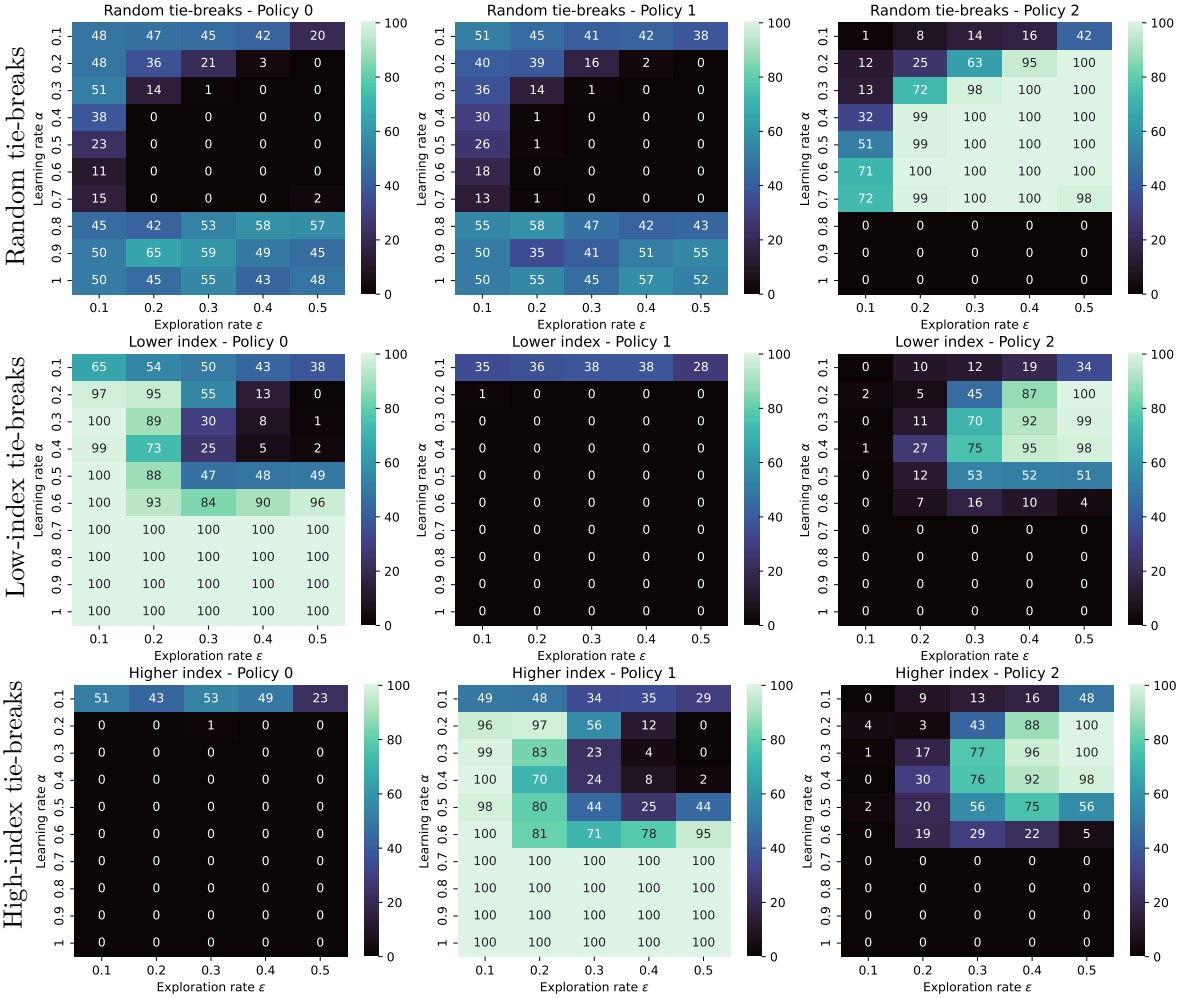

Figure 5: Heatmaps showing the frequency with which each policy was selected as the final greedy policy across 100 trials using different hyperparameter settings for each of the three different tie-breaking approaches: random (top row), lower-indexed action (middle row), and higher-indexed action (bottom row)

some degree of bias in the action selection, but presumably the user is indifferent to this bias (otherwise it would be represented in some way within the utility function).

To demonstrate the impact of value function interference, and the ability of non-random tie-breaking to mitigate its effects, empirical experiments were run on the MOMDP from Figure 4. These experiments used three variations of a tabular multi-objective Q-learning agent. The first variant is a baseline using the standard random approach to tie-breaking in the case of actions with equal scalarised values. The second variant deterministically breaks ties by favouring the action with the lower index (i.e. $a_1$ is preferred to $a_2$). As this could produce a bias towards the desired policies (both of which select $a_1$ in the first state), we also included a third variant which favours the action with the higher index.

Each agent's performance was evaluated over a hyperparameter sweep across different values for the learning rate ($\alpha$) and starting exploration parameter (using $\epsilon$-greedy exploration with $\epsilon$ linearly decayed to zero over the trial). The $\lambda$ and $\gamma$ hyperparameters were fixed at 0.95 and 1 respectively, and to produce more directed exploration Q-values were optimistically initialised to (12,0,0). For each set of hyperparameters 100 trials of each agent were executed. Each trial lasted for 500 episodes, and at the end of the trial the final greedy policy of the agent was identified.

Figure 5 is a heatmap showing the frequency of occurence of each of the possible policies for each agent/hyperparameter combination. Policy 3 has been omitted as it was never selected as the final policy in any trial.

### 2.3.1 Random tie-breaking

The results from the random tie-breaking agent (Figure 5, top row) clearly illustrate the problems caused by value function interference, as for large areas of the hyperparameter sweep the agents converged to the sub-optimal Policy 2 in the overwhelming majority of trials. Notably this was most likely to occur for trials where the learning rate $\alpha$ was in the middle of its range $0 \ldots 1$.

This can be explained by the nature of the utility function. The value learned for $Q(A, a_1)$ will over time converge to lie on the line segment between (7,-1,-5) and (7,-5,-1) (i.e. between the rewards for $B, a_1$ and $B, a_2$. The location along this line will depend on the frequency with which each action has been performed in state $B$ recently, as well as the magnitude of $\alpha$. Points on this line-segment will have the form $p = (7, -5 + 4x, -1 - 4x)$ where $0 \leq x \leq 1$. Action $a_1$ will be preferred to $a_2$ in state $A$ when $U(Q(A, a_1)) - U(Q(A, a_2)) > 0$. Substituting the definition of $p$ and the value of $Q(A, a_2)$ yields that this will occur when $16x^2 - 16x + 2 > 0$, which occurs when $x < (2 - \sqrt{(2)})/4$ and when $x > (2 + \sqrt{(2)})/4$ (i.e. when $Q(A, a_1)$ is close to either $Q(B, a_1)$ or $Q(B, a_2)$).

Consider the situation where $Q(A, a_1)$ is a good approximation to $Q(B, a_1)$, and now action $B, a_2$ is selected as a greedy action and executed. $Q(A, a_1)$ will be updated towards (7,-5,-1). A small value of $\alpha$ will produce a small change in $Q(A, a_1)$ meaning it will likely still be in the range where it is considered preferable to $Q(A, a_2)$. A large value of $\alpha$ will produce a large change in $Q(A, a_1)$, jumping over the portion of the line-segment where $A, a_2$ is preferred. Meanwhile middling values of $\alpha$ will result in $Q(A, a_1)$ falling roughly halfway between (7,-1,-5) and (7, -5, -1) (i.e. $x \approx 0.5$), which is the region where $A, a_2$ will be preferred. Hence it is this central range of $\alpha$ values where convergence to the incorrect policy is most likely to occur, which is evident in the experimental results.

There is also a tendency for sub-optimal convergence to become more frequent across a wider range of $\alpha$ values as the degree of exploration is increased. We note that the exploration values used here may seem high for such a simple MOMDP, but past research has shown that exploration rates may need to be higher for MORL than for single-objective RL (Vamplew et al., 2017a). Therefore this link between value function interference and high rates of exploration may prove to be problematic for more complex MOMDPs.

### 2.3.2 Deterministic tie-breaking

The results from the two deterministic tie-breaking agents (Figure 5, middle and bottom rows) demonstrate substantial improvements over the random tie-breaking agent (most notably for the case where $\alpha$=0.7, but more generally across the central portions of the hyperparameter search). Clearly the avoidance of random tie-breaking helps to mitigate, but not entirely avoid, the problems caused by value function interference – there are still combinations of hyperparameters where most/all trials converge to the incorrect final policy.

Comparing the results of the two deterministic tie-breaking agents, we see a clear effect of the process used to break ties – the first agent favours actions with a lower index, and so when it converges to one of the optimal policies, it has a clear preference for Policy 0 over Policy 1, whereas the second agent favours actions with a higher index and exhibits the opposite preference over these policies.

Given this capacity for the deterministic tie-breaking to induce different preferences over policies, we need to consider the possibility that the reduction in the frequency of convergence to sub-optimal policies is not due directly to the elimination of random tie-breaking, but instead due to an explicit bias away from Policy 2. In particular low-index tie-breaking will tend to favour $a_1$ in State $A$, which may in itself reduce the likelihood of convergence to Policy 2, regardless of any impact on value function interference. Figure 6 assists in differentiating between these causes by performing a pairwise comparison of the agents based on how frequently they incorrectly converge to Policy 2. It can be seen from Figure 6(middle) that the low-index deterministic tie-breaking agent converges to this policy with lower or comparable frequency to the random tie-breaking agent for every hyperparameter setting which was tested. Overall it selects this policy in 1493

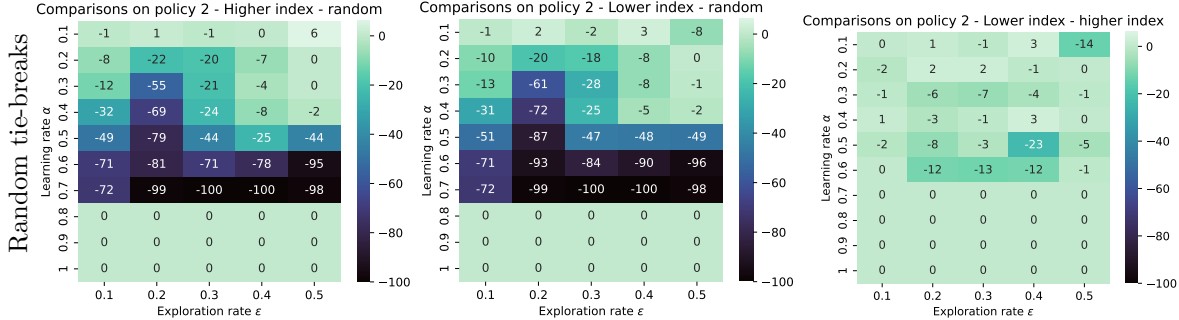

Figure 6: Heatmaps showing the difference in the frequency with which the incorrect Policy 2 was selected as the final greedy policy across 100 trials using different hyperparameter settings for each combination of the three different tie-breaking approaches: Higher-index - Random (left); Lower-index - Random (middle); Lower-index - Higher-index (right)

fewer trials than the random tie-breaking agent (almost 30% of the total number of trials). The high-index deterministic tie-breaking agent also improves substantially over the random tie-breaking agent, selecting the sub-optimal policy in 1385 fewer cases (Figure 6 (left)), but this is 108 cases worse than the low-index agent. So while the low-index agent's bias away from action $a_2$ in state $A$ is partially responsible for the reduction in trials converging to Policy 2, this accounts for just $108/1493$ cases – more than 90% of the improvement is due to avoiding the stochasticity of the random tie-breaker rather than the specific bias of the deterministic tie-breaker.

These results demonstrate that deterministic tie-breaking may be beneficial in the case of MOMDPs where some actions share identical utility.

## 2.4 Interference due to changes in greedy action

The results in the previous section demonstrate that non-random tie-breaking can alleviate the impact of value function interference. However it fails to eradicate this problem, as there are still a considerable number of cases where the agent converges to the non-optimal Policy 2 (including some parameter settings where this occurs in all trials). Therefore there must be a secondary cause of value function interference within this deterministic MOMDP.

We hypothesise that this may occur in cases where the agent's choice of greedy action changes. In a deterministic MOMDP this could arise due to optimistic initialisation of the Q-values – each action will need to be sampled multiple times before its associated Q-value estimates converge closely enough to their true values for the optimality (or otherwise) of that action to be accurately established. The number of samples required will depend on the learning rate, and also on the size of the gap in utility between each action and the optimal action.

Consider a variant of the MOMDP from Figure 4, in which the reward vector for $a_2$ in state B is $(7-\delta, -5, -1)$ meaning that its utility is now lower than that of action $a_1$ by $2\delta$. This change means that Policy 0 $(a_1,a_1)$ is now the sole optimal policy. The magnitude of $\delta$ will inversely influence the number of samples and Q-value updates required to establish that $U(Q(B, a_1)) > U(Q(B, a_2))$, and hence will reduce the number of occasions on which the agent switches its choice of greedy action for state $B$. Note that this change to the environment has no impact on the relative utility of Policy 0 $(a_1,a_1)$ and Policy 2 $(a_2,a_1)$ and so, absent the effects of value function interference, changing the value of $\delta$ should have no effect on how frequently Policy 2 is selected by the agent.

However the results in Figure 7 show that even a tiny $\delta$ value of 0.0001 is sufficient to allow the agent to converge to the optimal policy (Policy 0) in almost all trials across all values of the learning hyperparameters. A larger $\delta$ value is sufficient for all trials to converge successfully to this optimal policy.

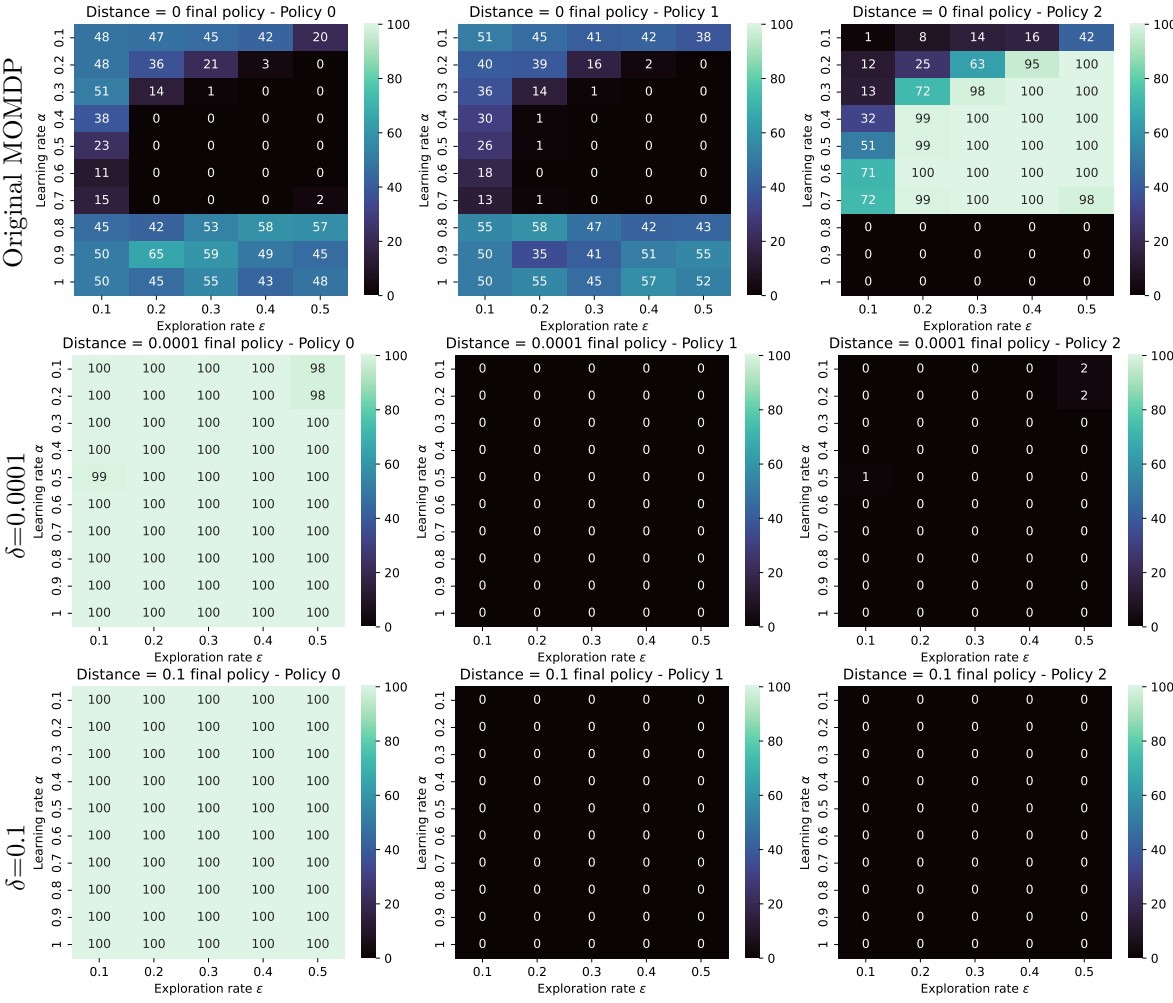

Figure 7: Heatmaps showing the frequency with which the correct Policy 0 and incorrect Policies 1 and 2 were selected as the final greedy policy for the deterministic MOMDP across 100 trials using different hyperparameter settings. The top row corresponds to the original MOMDP from Figure 4, while the other rows use a modified reward for action $a_2$ in state $B$, where the first objective has been reduced by $\delta$. These experiments use the random tie-breaker.

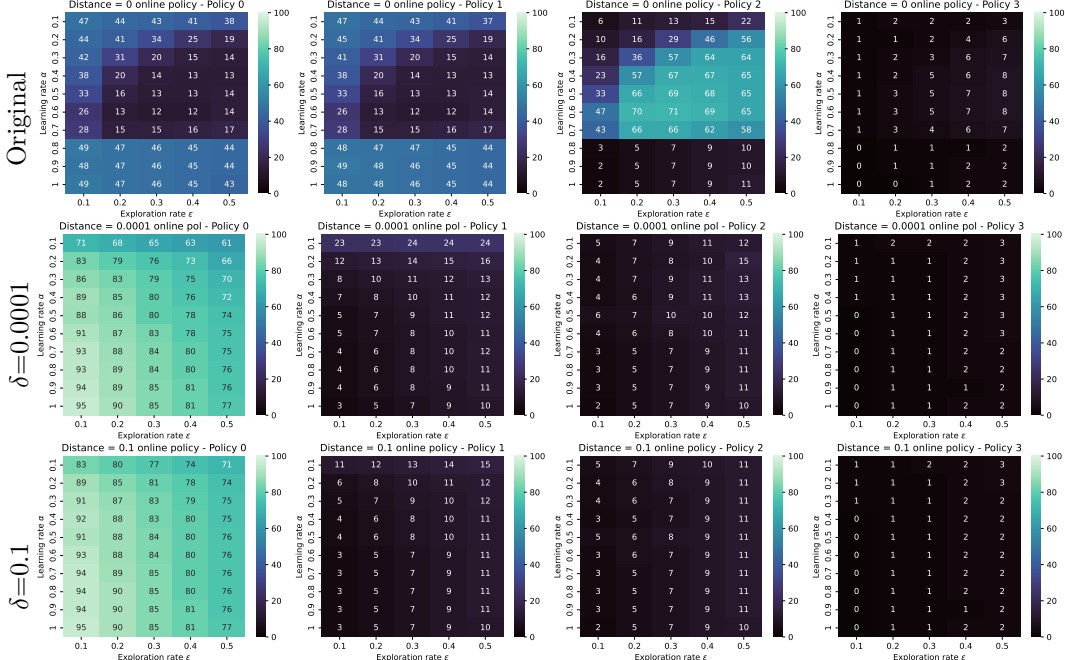

Figure 8: Heatmaps showing the frequency with which each of the policies were selected during learning, as a percentage of all episodes over all 100 trials. The top row corresponds to the original MOMDP from Figure 4, while the other rows use a modified reward for action $a_2$ in state $B$, where the first objective has been reduced by $\delta$. These experiments use the random tie-breaker.

Further insight into the effect of $\delta$ on the prevalence of value function interference can be gained by looking at the selection of actions during learning rather than just in the final policy. Figure 8 shows the percentage of episodes in which each policy is executed during learning (including episodes in which exploratory actions are selected). It can be seen that as the magnitude of $\delta$ increases, the agent increasingly favours the optimal Policy 0 during training. Significantly, this is due not just to less frequent execution of Policy 1 (whose utility is directly effected by $\delta$) but also by reduced execution of Policies 2 and 3 – this provides clear evidence that value function interference must be arising in the estimated value of $Q(A, a_1)$ for smaller values of $\delta$.

## 2.5 Value function interference – summary

The results reported in this section establish the existence of the learning phenomenon that we have named *value function interference*. This arises when the expected vector returns learned by an RL agent are inadequate to effectively determine the actions and policies which are optimal with respect to the utility function. This can significantly hinder learning, and may lead to convergence to a sub-optimal policy.

The most obvious situation in which value function interference can occur is where the environment itself is stochastic with regards to either rewards and/or state transitions. However we have demonstrated that interference can also occur in deterministic environments, either due to stochasticity in tie-breaking of greedy actions, or due to frequent changes in the choice of greedy action (i.e. as might be induced by optimistic initialisation). The latter issue may be especially problematic in the context of value-based deep RL. Schaul et al. (2022) identified extremely high rates of change in the greedy policy in value-based deep RL algorithms such as DQN and R2D2, which suggests that multi-objective extensions of those algorithms may be particularly susceptible to value-function interference.

The experiments in this section demonstrated value function interference arising even in the context of otherwise stable learning algorithms based on tabular representation of Q-values. In practice for more complex environments the Q-value estimates may themselves be noisy due to the need to use function approximation. These errors in estimates may result in changes in the action which is identified as greedy

in each state, leading to variations over time in the vector value updates for prior states, which could result in increased levels of value function interference. This is particularly likely to occur for highly non-linear definitions of utility as small variations in the estimated vector values can lead to large variations in the estimated utility; for example, unstable and slow learning has previously been reported when using a thresholded lexicographic approach to utility on stochastic problems (Vamplew et al., 2021b;a; Ding et al., 2025).

## 3 Sensitivity of MORL to overestimation

It is well established that off-policy methods such as Q-learning are prone to overestimation of Q-values when used in conjunction with function approximation, and there has been considerable effort at addressing this issue (Anschel et al., 2017; Van Hasselt et al., 2018; Ly et al., 2024). Li (2023) notes that for single-objective RL, overestimation may not actually prevent the optimal policy from being learned as "if all values are uniformly higher, relative action preferences are still preserve". More formally:

$$Q(s, a^*) > Q(s, a) \implies Q(s, a^*) + \delta > Q(s, a) + \delta \tag{3}$$

While this would also apply to MORL with linear scalarisation, it is not necessarily true if non-linear scalarisation is used (Giuseppi & Pietrabissa, 2020; Mabsout et al., 2025). For a non-linear utility function, even a small amount of overestimation in the Q-value vectors may be sufficient to change the preference ordering of actions, even if that overestimation is consistent over all actions:

$$U(Q(s, a^*)) > U(Q(s, a)) \;\not\!\!\!\implies\; U(Q(s, a^*) + \delta) > U(Q(s, a) + \delta) \tag{4}$$

In the remainder of this section, we present a brief empirical demonstration of the impact which overestimation, consistent or otherwise, might have on the behaviour of a value-based MORL agent.

### 3.1 Design of overestimation experiments

The benchmark environment for our experiments with overestimation is the DST-Mixed task from Vamplew et al. (2015) (Figure 9). This is a small grid-world where the agent controls the movements of a submarine searching for undersea treasure. There are ten Pareto-optimal deterministic policies representing different trade-offs between the time and treasure objectives. The Pareto front contains several concavities – as a result only four of these policies can be optimal under linear scalarisation.

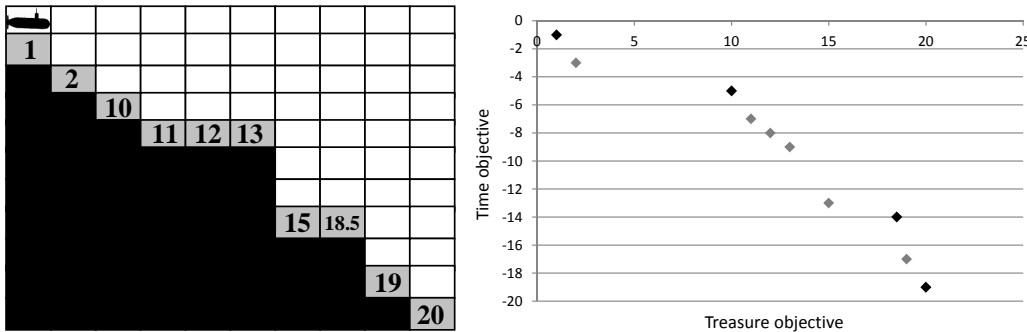

Figure 9: A visualisation of the DST-Mixed environment (left), and its Pareto front (right). Black points show policies which can be found by linear scalarisation, while grey points are policies which can only be found via non-linear scalarisation. (Reproduced from Vamplew et al. (2015))

For simplicity we map this environment to a multi-objective multi-armed bandit (MOMAB), where each arm of the bandit corresponds to one of the Pareto-optimal policies. This allows us to avoid the additional complexities arising from variations in overestimation at different states within a trajectory. We also omit

the process of learning Q-values. Instead we focus on the impact of overestimation of the Q-values for the starting state on the choice of policy to be executed, and how this varies depending on the nature of the utility function. Three utility functions were considered – linear scalarisation, thresholded lexicographic ordering (TLO), and Chebyshev distance. The latter uses a utopian reference point $z^*$. In the MORL literature, this is typically derived from the maximum return received for each objective during learning (i.e. this value is updated during the learning process) (Van Moffaert et al., 2013; Qin et al., 2020; Kim et al., 2022). As these experiments were not simulating the learning process itself, this point was set to the maximum value of each objective under any policy, plus a small positive offset so $z^* = (20.1, -0.9)$.

$$U_{linear}(r_{treasure}, r_{time}) = w_1 r_{treasure} + w_2 r_{time} \tag{5}$$

$$U_{TLO}(r_{treasure}, r_{time}) = \begin{cases} r_{treasure}, & \text{if } r_{time} \geq \text{threshold} \\ r_{time}, & \text{otherwise} \end{cases} \tag{6}$$

$$U_{chebyshev}(r_{treasure}, r_{time}) = -\max(w_1|z_1^* - r_{treasure}|, w_2|z_2^* - r_{time}|) \tag{7}$$

| Utility function U | Parameters | Range |
|---|---|---|
| Linear | Objective weights $w_1, w_2$ | $w_1$ in $[0.17 \ldots 0.65]$, $w_2 = 1 - w_1$ |
| TLO | Threshold for time | $[-20 \ldots 0]$ |
| Chebyshev distance | Objective weights $w_1, w_2$ | $w_1$ in $[0.01 \ldots 0.99]$, $w_2 = 1 - w_1$ |

Table 2: The ranges used for the utility function parameters in the overestimation experiments. Values were sampled uniformly within these ranges.

Prior to each arm-pull the following process was performed:

- A random set of parameters were generated for the utility function. These were drawn uniformly from the ranges specified for each utility function in Table 2. These ranges were chosen as they give rise to an approximately even distribution over the policies when applied to the actual Q-values.

- Positive noise was temporarily added to the true Q-values for each arm to simulate the effect of overestimation.

- The arm to pull was selected greedily based on these 'overestimated' Q-values.

This process was repeated 1000 times for each utility function. The impact of the overestimation was tracked via the following statistics:

- The number of pulls for which a non-optimal arm was selected.

- The mean normalised regret incurred. For any arm pull, the regret can be calculated in terms of the loss in utility incurred by selecting this arm rather than the optimal arm

$$regret(a) = E[U(r_{a^*})] - E[U(r_a)], \text{ where } a^* \text{ is the optimal arm} \tag{8}$$

However the range of the utility varies substantially between the three different utility functions, as can be seen in the colorbars in Figure 11, so a direct comparison of regret is not appropriate. Therefore we use a *worst-case normalised* regret measure, where the regret from the performed action is normalised with respect to the regret of the worst-possible action. This gives a value in the range $0 \ldots 1$, enabling comparison across utility functions with different ranges.

$$regret_{WCN}(a) = regret(a)/regret(a^{worst}), \text{ where } a^{worst} \text{ is the least-optimal arm} \tag{9}$$

The entire experimental process was repeated twice for two different types of overestimation. In the *consistent overestimation* trials the same overestimation value $\delta$ was added on to both objectives for all Q-values. In the *inconsistent overestimation* trials for each arm a different overestimation value was uniformly sampled from the range $0 \ldots \delta$ for each objective. In both cases multiple experiments were run for values of $\delta$ between 0 and 5.

### 3.2 Overestimation experiments results and discussion

Figure 10 illustrates the impact of consistent overestimation. As expected, this has no impact on the performance of an agent using linear scalarisation, regardless of the magnitude of the overestimation. The Q-values of all arms are increased by the same amount which preserves their ordering. In contrast the highly non-linear nature of TLO makes it very sensitive to overestimation. Even a minimal level of overestimation is sufficient to cause a substantial number of incorrect decisions. The number of errors increases rapidly as the magnitude of overestimation increases, until almost all decisions are incorrect. Meanwhile the other non-linear utility function Chebyshev distance performs somewhere between these extremes. Examining the normalised regret metric, it can be seen that not only is TLO prone to a higher frequency of errors than Chebyshev, but the impact of those errors (in terms of lost utility) is also much greater.

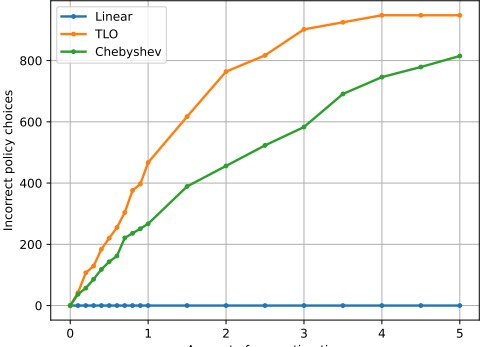 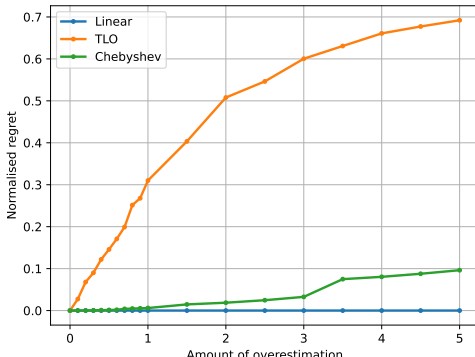

Figure 10: Results for the DST-Mixed environment for various utility functions with **consistent** overestimation of Q-values. The left plot shows the number of incorrect policy choices over 1000 trials, while the right plot shows the mean normalised loss of utility.

The utility contour plots in Figure 11 provide insight into the reason for these results. It can be seen that the gradient of the linear utility function is constant across the entire objective space. Therefore adding a constant level of overestimation to all Q-values increases the perceived utility of all policies equally and so has no impact on action selection. The Cheybshev function has a similarly constant gradient for large parts of the objective space, but the use of the *max* operator in conjunction with objective weights can result in inconsistent gradients. For example in the case shown in Figure 11, the optimal policy based on the true Q-values is the 7th from the left (#7), whereas once consistent noise is added the selected policy will be the 6th from the left (#6). Meanwhile the TLO utility function exhibits a uniform gradient if time exceeds the threshold, but a discontinuity at that threshold. Therefore adding constant overestimation results in varying changes in utility between the different policies – in particular the policy with true return (18.5, -14) undergoes a marked increase in utility when the overestimation is added, jumping from the 8th-ranked policy to the 1st-ranked. Where a policy's return lies just below the threshold value, even a small amount of overestimation can result in incorrect action selection and a large loss in utility.

The results of the experiments using inconsistent overestimation are shown in Figure 12. For all three utility functions, inconsistent overestimation is more disruptive than consistent overestimation.[5] In this

---

[5]When comparing Figures 10 and 12, keep in mind that for the former the level of overestimation is always equal to $\delta$, whereas for the latter the overestimation lies between 0 and $\delta$ and so will average $\delta/2$. So, for example, the most appropriate comparison for results at $\delta = 0.2$ in Figure 10 are the results for $\delta = 0.4$ in Figure 12.

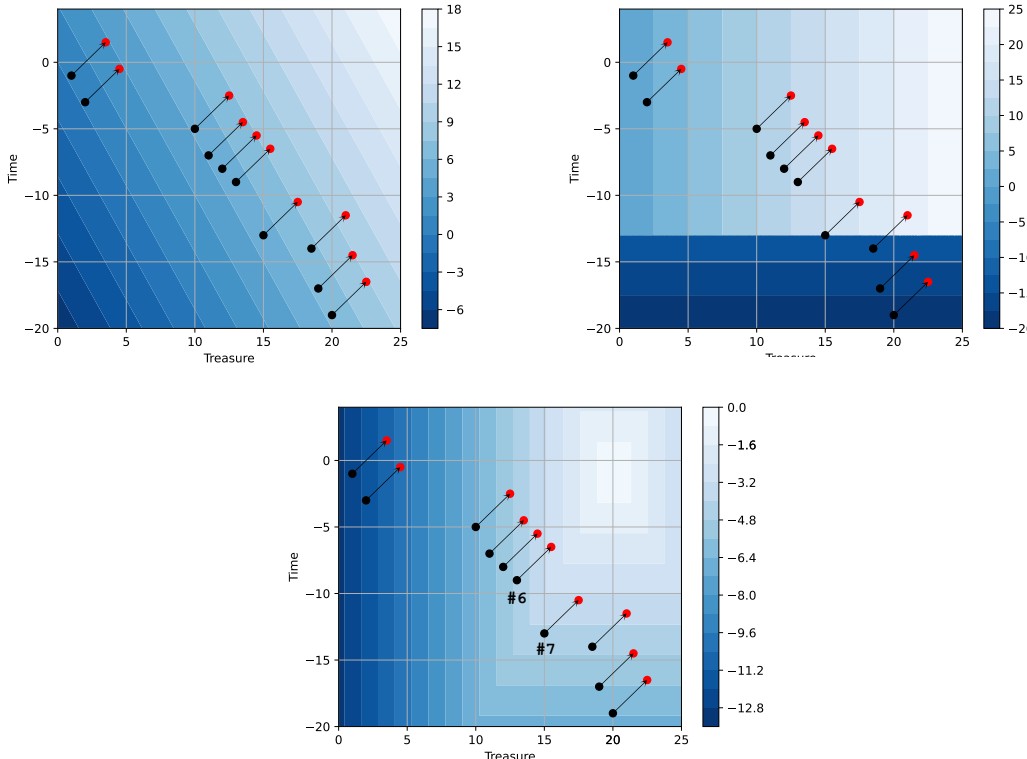

Figure 11: Contour plots mapping treasure/time to utility for a representative choice of each utility function: (top left) linear scalarisation with $w_1 = 0.65, w_2 = 0.35$, (top right) TLO with a time threshold of -13, and (bottom) Chebyshev distance with $w_1 = 0.65, w_2 = 0.35$. Black dots show the true return for the Pareto-front of policies. Arrows and red dots show the addition of consistent noise of +2.5 added to those true returns to represent over-estimated Q-values.

case, linear scalarisation is also affected as the variations in overestimation between different arms can cause the preferences between those arms to change. Nevertheless it still remains considerably more robust than TLO, both in terms of the rate of errors and the utility lost due to those errors. Interestingly while Chebyshev distance is less robust than linear scalarisation with regards to the number of incorrect policy selections, it does seem to be slightly more robust with regards to normalised regret for larger magnitudes of inconsistent overestimation.

## 4 Conclusion

This paper has identified two previously unreported issues which may impact on the performance of value-based MORL algorithms, and provided empirical evidence of this impact on tabular multi-objective Q-learning when applied in combination with a non-linear utility function:

- Value function interference is an issue arising from the use of vectors to represent the expected future multi-objective return. In some cases this is not sufficient information to establish the optimal policy. This is particularly evident when trying to learn ESR-optimal policies for stochastic environments, but this paper has also demonstrated that it is possible for value function interference to arise in the context of SER-optimal policies for deterministic environments. We have shown empirically that in the latter case, modifying the core Q-learning algorithm to use non-random tie-breaking when two actions have equivalent utility can ameliorate, but not totally eliminate, the impact of value function interference.

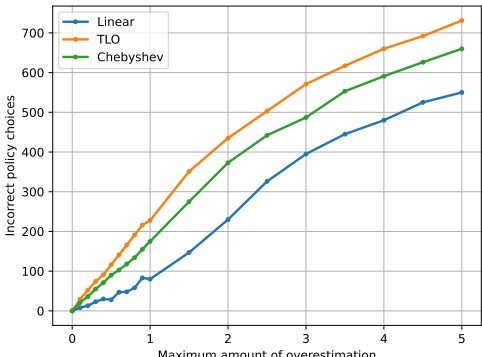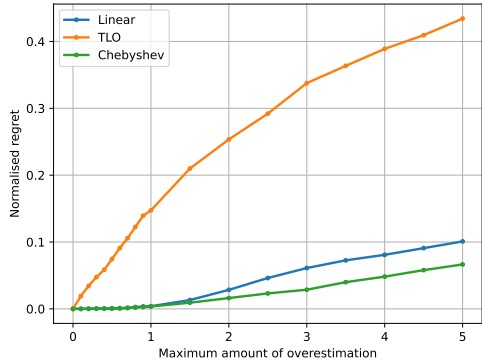

Figure 12: Results for the DST-Mixed environment for various utility functions with **inconsistent** overestimation of Q-values. The left plot shows the number of incorrect policy choices over 1000 trials, while the right plot shows the mean normalised loss of utility.

- It is well established that value-based RL algorithms can be prone to overestimation bias when used in conjunction with function approximation. While this problem is not unique to MORL, the results in Section 3 highlight that MORL methods may be particularly sensitive to overestimated Q-values when applied with non-linear utility functions.

We suggest that one promising course of action to address value function interference is to incorporate concepts from distributional RL into MORL, as pioneered by Hayes et al. (2021),Reymond et al. (2023) and Röpke et al. (2023). If the agent learns a distribution over the vector returns for each state-action, then this can be combined with the utility function so as to estimate the ESR.

For example for the MOMAB in Figure 2 assume the agent has correctly learned that the distribution of future returns for action $a_1$ is (7,-1, -5) with $p = 0.5$ and (7,-5, -1) with $p = 0.5$. From this information the agent can correctly estimate the ESR utility of $a_1$ as the probabilistically-weighted sum of those return vectors.

$$\begin{aligned}
U_{ESR}(a_1) &= 0.5U(7, -1, -5) + 0.5U(7, -5, -1) \\
&= 0.5 * 9 + 0.5 * 9 \\
&= 9
\end{aligned} \tag{10}$$

Another possible solution is to adopt the approach of Cai et al. (2023). This involves a form of potential-based reward shaping derived by using the multi-objective utility function to scalarise the accumulated vector return on each time-step, and using a conventional RL algorithm to learn a policy from those scalar rewards. This means the agent is directly learning the ESR value, and so eliminates the possibility of value-function interference. As it is learning a scalar return, this approach may also mitigate against the sensitivity of non-linear utilities to overestimation in Q-values. However it no longer supports some of the benefits of MORL, such as multi-policy learning, or the increased explainability offered by vector Q-values (Zhan & Cao, 2019; Dazeley et al., 2023).

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
