# OpenReview forum: "Issues with Value-Based Multi-objective Reinforcement Learning: Value Function Interference and Overestimation Sensitivity"
_TMLR — Rejected by TMLR_

### Review · Reviewer_UWsj · 2026-02-24

**Summary Of Contributions:**

The paper analyzes value-based multi-objective RL built on vector-valued Q-learning with greedy action selection through (potentially non-linear) utility/scalarisation, and argues that two underappreciated failure modes can materially degrade learning: value function interference, where learning only the expected return vector can induce systematically wrong greedy choices under non-linear utilities (especially for ESR in stochastic settings, but also in deterministic tasks via tie-breaking and policy-switching effects). Meanwhile, it heightened sensitivity to Q-value overestimation, where even seemingly benign estimation bias can flip action rankings under non-linear utilities, which is motivated by the well-known overestimation tendency in Q-learning with approximation.

strengths: the mechanisms are explained with minimal, inspectable examples and backed by clear tabular/bandit-style experiments plus concrete mitigations like deterministic tie-breaking and a discussion of distributional remedies.

weaknesses: evidence is mostly toy/tabular or bandit-reduced, so it remains unclear how strongly these effects persist under function approximation at scale, and the proposed mitigation is partial/ad hoc rather than a fully ESR-aligned solution.

**Additional Comments:**

## References

- [1] A Survey of Multi-Objective Sequential Decision-Making. Roijers et al. 2013.
- [2] Actor-Critic Multi-Objective Reinforcement Learning for Non-Linear Utility Functions. Reymond et al. 2023.
- [3] Distributional Pareto-Optimal Multi-Objective Reinforcement Learning. Cai et al. 2023.
- [4] Distributional Multi-Objective Decision Making. Röpke et al. 2023.
- [5] A Distributional Perspective on Reinforcement Learning. Bellemare et al. 2017.
- [6] Deep Reinforcement Learning with Double Q-Learning. van Hasselt et al. 2016.
- [7] Issues in Using Function Approximation for Reinforcement Learning. Thrun & Schwartz 1993.
- [8] The impact of environmental stochasticity on value-based multiobjective reinforcement learning. Vamplew et al. 2022.

**Audience:**

Yes

**Audience Explanation:**

Many practitioners still treat “vector Q + non-linear scalarisation” as a natural baseline, and this paper clarifies where that baseline can be structurally misaligned with ESR objectives [1,8]. While in this submission, the discussion connects directly to known instability in value-based deep RL where the greedy policy changes rapidly over training.

**Claims And Evidence:**

No

**Claims Explanation:**

The paper convincingly shows **existence** of the phenomena, but the evidence does not yet support broader claims about typical impact in realistic MORL with function approximation. Instead, the overestimation study uses synthetic noise added to fixed Q-vectors, which is informative but not clearly representative of the structure of estimation errors in learned value functions. Besides, several conclusions depend strongly on particular exploration/initialization choices in very small MDPs, and the paper lacks a stronger separation between general principles and setting-specific effects.

**Requested Changes:**

- Reframe novelty to clearly separate what is new from established SER/ESR distinctions and prior discussions of ESR needing more than expected vectors [1,8].
- Add experiments with **function approximation** (at least a small deep value-based MORL baseline) to test whether interference and overestimation sensitivity persist beyond tabular settings.
- Compare against at least one **principled ESR-aligned** alternative (e.g., distributional MORL / utility-critic style methods) rather than only tie-breaking heuristics [2,3,4,5].
- Non-critical: Provide a more formal characterization of “value function interference,” including conditions where it cannot change greedy action selection (beyond “linear utility”).
- Non-critical: Expand the overestimation study to include correlated/state-dependent noise models and relate more directly to standard overestimation analyses in value-based RL [6,7].
- Besides, release code.

---

> ### Author Response · Authors · 2026-04-14
>
> The feedback from both Reviewer UWsj and HT9F has indicated that our discussion of the distinction between ESR/SER missed some key points. If permitted to submit a revised paper, we will extend that discussion in the Introduction to clarify the relationship between these optimisation criteria and Algorithm 1, as well as more thoroughly discussing the implications of our findings – as noted in our response to reviewer HT9F, our results when combined with those of [1,8] and Ding et al (2025) indicate that vector-valued Q-learning is potentially unreliable for non-linear utility regardless of whether ESR or SER optimality is required.
>
> Regarding the requested change to add experiments with function approximation, we agree that including experiments on a more complex environment would strengthen our paper. We did carry out an extensive set of experiments applying single-objective DQN and a non-linear MODQN algorithm to the MOLunarLander environment from the MO-Gymnasium library. However the results of both the algorithms were so noisy that it was difficult to isolate the existence of, or effect of, VFI or overestimation – even the single-objective DQN algorithm produced results which varied substantially between multiple seeds using the same linear scalarisation of rewards. If required, we could redo these experiments on a simpler environment such as the MinecartDeterministic environment.
>
> We can extend our existing tabular experiments (and, if required the additional Deep RL experiment) to include a principled ESR-aligned algorithm. Our preferred choice would be the approach of Cai et al [3] as we intend to work with that algorithm in our future research in any case.
>
> We have had discussions amongst our group about whether we can “Provide a more formal characterization of value function interference, including conditions where it cannot change greedy action selection (beyond “linear utility”).” While we agree that such a characterization would be useful, we don’t see a path to a more formal specification than we were able to provide in the introduction to Section 2.
>
> We appreciate the reviewer’s point regarding the limitations of our overestimation study (ie not considering correlated or state-dependent noise models). We note that our intention was primarily to demonstrate the potential for increased impact of overestimation when combined with a non-linear utility rather than aiming to accurately simulate the nature of noisy estimates that might arise during actual learning. If we do carry out an extended experiment using function approximation, we will adopt methods for measuring overestimation as used in the prior literature [6, 7]. We also note that shortly after submitting this paper we became aware of one very recently released paper which addresses the issue of estimation bias in the context of successor functions and MORL (although only under linear utility), so we would add discussion of that to a revised submission: Lu, M., Zhou, Y., Xie, S., Peng, Y., Zhang, X., & Zhang, Y. (2026). Tackling value estimation bias in successor features by distributional reinforcement learning. Applied Soft Computing, 114814.
>
> We are happy to release the code used in this study – we will clean up and document the code and set up a github to share it publicly.

---

### Review · Reviewer_HT9F · 2026-03-15

**Summary Of Contributions:**

This paper studies two issues raised by the authors in multi objective reinforcement learning (MORL): (1) value function interference and (2) value overestimation. Value function interference refers to learning suboptimal or incorrect policies when applying vector Q learning to MORL with a utility function. Value overestimation is a known issue in RL and the utility function in MORL exacerbates the issue. The authors conducted a series of ablations illustrating both issues. For issue (1), they propose non-random action tie braking which improves the chance of identifying the optimal policy.

**Additional Comments:**

* Other than that I think the premise of the paper is potentially flawed, I think the analyses and explanations are valid and reasonable. The final comment on learning return distributions makes sense.
* The chosen experiment environment are a bit too simple, demonstrating this in a more realistic setting would be very useful.

**Audience:**

No

**Audience Explanation:**

Given I think the premise of this paper is potentially flawed, I am not able to assess this until further clarification from the authors.

**Claims And Evidence:**

No

**Claims Explanation:**

This paper claims value interference and overestimation are significant issues in MORL. Which their ablation results provide evidence for these claims, I don't think the overall problem is set up clearly enough to assess these claims. My main question is the following:

It seems like the paper, especially the value interference section, revolves around the idea of using an SER algorithm (i.e., algorithm 1) to optimize for an ESR criterion. This is what the value interference definition and all subsequent analyses are based on if I understood correctly. But I think this approach is fundamentally flawed?

**Requested Changes:**

* My main request to the authors is to clarify the premise of the paper, i.e., whether they focus on using SER algorithms for ESR optimization, and if so why should they do this? If this is a standard practice in MORL and the authors' goal is to critique this, then this motivation should be made very clear early on.
* Could you explain why algorithm 1 uses eligibility trace? It's not clear to me that this is a crucial feature in your analysis.

---

> ### Author Response · Authors · 2026-04-14
>
> The reviewer’s observation that the MO Q-learning algorithm (Algo 1) is a more natural fit for SER than ESR optimisation is correct, and on re-reading our paper we realise we should have addressed this issue more directly.
>
> We would argue that the work we are reporting is still valid and of interest to TMLR’s audience for the following reasons:
> -	For the deterministic MDP used in Section 2.2, the same deterministic policy is optimal under both the SER and ESR criteria. Therefore the forms of value function interference observed in these experiments (due to random tie-breaking, and the effects of changes in greedy action during learning) can arise even for SER learning.
> -	Prior work by Ding et al (2026) and Vamplew et al (2021a) has shown that Algorithm 1 can perform very poorly in terms of SER-optimality even on very simple MDPs. Those results in conjunction with ours showing poor performance in terms of ESR indicate that this form of vector-valued Q-learning is actually reliable only in the context of linear utility. This is a point which we would emphasise more directly in a revision of this paper.
> -	As noted by Reviewer UWsj “Many practitioners still treat vector Q + non-linear scalarisation’ as a natural baseline”, and apply it even in the context of ESR optimisation (in fact, our experience is that a considerable proportion of MORL research fails to even identify whether it is performing ESR or SER optimisation). Hence highlighting that this approach “can be structurally misaligned with ESR objectives” is a worthwhile contribution.
>
> Regarding the use of eligibility traces in Algorithm 1, the reviewer is correct that this is not a crucial feature of our analysis. The codebase which we used for our experiments used eligibility traces, and so we included them in the algorithm for the sake of consistency. We can add a footnote to clarify this.
>
> We agree that the experiment environments are simple. This was a deliberate design choice on our part so as to demonstrate the existence of these issues within a setting that was easy to understand and analyse. Please see our response to Reviewer UWsj for a more detailed discussion of the possible extension of our work to a more realistic setting involving the use of function approximation rather than tabular RL.
>
> References:
>
> Kewen Ding, Peter Vamplew, Cameron Foale, and Richard Dazeley. An empirical investigation of value-based multi-objective reinforcement learning for stochastic environments. The Knowledge Engineering Review, 40:e6, 2025.
>
> Peter Vamplew, Cameron Foale, and Richard Dazeley. The impact of environmental stochasticity on valuebased multiobjective reinforcement learning. Neural Computing and Applications, pp. 1–17, 2021a.

---

> > ### Comment · Reviewer_HT9F · 2026-04-26
> >
> > Thank the authors for the clarification. I believe the premise of the paper is quite clear now and I believe the paper would be of interest to the TMLR audience.
> >
> > I agree with reviewer UWsj that adding an experiment with function approximation would substantially strengthen the claims. I understand that the noisiness in your preliminary experiment makes it difficult to disentangle VFI and overestimation. Since the focus is on function approximation, one idea could be to remove exploration from the confounding factors and sample directly from the known state space, e.g., using env.observation_space.sample().

---

### Review · Reviewer_Jy96 · 2026-04-13

**Summary Of Contributions:**

**Summary**

This paper conducts an empirical investigation and presents a discussion on potential solutions to the challenges of value function interference and sensitivity to overestimation within the context of a single agent multi-objective reinforcement learning framework employing a non-linear utility function.


**Reproducibility**

The source code for the experiments was not published.


**Limitations**

1. Although the paper introduces candidate solutions to mitigate value function interference, it lacks proposed solutions for the issue of value overestimation sensitivity. It would be beneficial for the authors to elaborate on any potential strategies or insights they might have regarding this aspect.
2. Regarding the value overestimation problem characterized by inconsistent overestimation, the paper could benefit from a more detailed explanation of why the normalized regret for Chebyshev distance outperforms the Linear Scalarization method. This clarification would enhance the reader’s understanding of the comparative advantages of these approaches.
3. It would be useful to publish the source code for the experiments, to support and accelerate further research in the area.

**Audience:**

Yes

**Audience Explanation:**

The paper shed light on a relevant RL problem that is relevant for the general community, beyond multi-objective RL.

**Claims And Evidence:**

Yes

**Claims Explanation:**

The empirical results from the multi-objective armed bandit (MOMAB) task (for value interference) and DST-Mixed task (for value overestimation) support the claims.

**Requested Changes:**

1. The X-axis label for the second plot of Figure 11 appears to be partially obscured, likely due to image cropping. It is recommended to ensure that all labels are fully visible to maintain clarity and comprehensibility of the figures.

---

> ### Author Response · Authors · 2026-04-14
>
> We are happy to release the code used in this study – we will clean up and document the code and set up a github to share it publicly.
>
> Our intention was not to solve the overestimation problem, but rather to highlight the increased disruptive potential of overestimation when used in conjunction with a non-linear utility function. We did note in the Conclusion that approaches like that of Cai et al which map the multi-objective problem back to learning of a scalar value could potentially mitigate this effect, so we could expand on that discussion id required.
>
> We will re-examine the results for the normalized regret for Chebyshev distance vs Linear scalarization and try to provide more insight into this result.
>
> Thank you for catching the issue with the x-axis on Figure 11. We will revisit our original graphs and crop them more carefully to fix this.

---

### Decision · Action_Editor_WRYq · 2026-06-01

**Recommendation:** Reject

**Audience:**

Yes

**Audience Explanation:**

All reviewers agree that the paper considers a relevant and important problem that is of great interest to TMLR audience.

**Claims And Evidence:**

No

**Claims Explanation:**

The reviewers agree that the paper addresses an important problem; however, some reviewers find that certain claims are not supported by sufficient evidence. In particular, they note that the results are limited to toy, tabular, and bandit-style settings, and do not convincingly demonstrate that the observed issues extend to broader settings. As a result, they find that some aspects of the work do not appear to be sufficiently validated by the presented experiments.

**Resubmission Of Major Revision:**

The authors may consider submitting a major revision at a later time.